# Trends in inequalities in Children Looked After in England between 2004 and 2019: a local area ecological analysis

Davara Lee Bennett ![ORCID], Kate E Mason ![ORCID], Daniela K Schlüter, S Wickham, Eric TC Lai, Alexandros Alexiou, Ben Barr, David Taylor-Robinson

► Prepublication history and additional materials for this paper is available online. To view these files, please visit the journal online (http://dx.doi.org/10.1136/bmjopen-2020-041774).

BB and DT-R contributed equally.

Department of Public Health, Policy and Systems, Institute of Population Health, University of Liverpool, Liverpool, UK

**Correspondence to**
Ms Davara Lee Bennett; davara.bennett@liverpool.ac.uk

## ABSTRACT

**Objective** To assess trends in inequalities in Children Looked After (CLA) in England between 2004 and 2019, after controlling for unemployment, a marker of recession and risk factor for child maltreatment.

**Design** Longitudinal local area ecological analysis.

**Setting** 150 English upper-tier local authorities.

**Participants** Children under the age of 18 years.

**Primary outcome measure** The annual age-standardised rate of children starting to be looked after (CLA rate) across English local authorities, grouped into quintiles based on their level of income deprivation. Slope indices of inequality were estimated using longitudinal segmented mixed-effects models, controlling for unemployment.

**Results** Since 2008, there has been a precipitous rise in CLA rates and a marked widening of inequalities. Unemployment was associated with rising CLA rates: for each percentage point increase in unemployment rate, an estimated additional 9 children per 100 000 per year (95% CI 6 to 11) became looked after the following year. However, inequalities increased independently of the effect of unemployment. Between 2007 and 2019, after controlling for unemployment, the gap between the most and least deprived areas increased by 15 children per 100 000 per year (95% CI 4 to 26) relative to the 2004–2006 trend.

**Conclusions** The dramatic increase in the rate of children starting to be looked after has been greater in poorer areas and in areas more deeply affected by recession. But trends in unemployment do not explain the decade-long rise in inequalities, suggesting that other socioeconomic factors, including rising child poverty and reduced spending on children's services, may be fuelling inequalities. Policies to safely reduce the CLA rate should urgently address the social determinants of child health and well-being.

### Strengths and limitations of this study

► This study is the first to quantify inequalities in child welfare outcomes in England longitudinally, using segmented mixed-effects models to show that the gap in rates of children becoming looked after between the most and least deprived areas is on the rise after controlling for unemployment.

► The study uses routinely available data for the whole of England and explores several child welfare outcomes to describe trends throughout the child welfare system.

► An important limitation is that, using an ecological area-level analysis, we cannot conclude that children becoming looked after were directly affected by the exposures of interest.

forms of acute adversity, may differ markedly from those of their peers. On average, individuals who have been looked after face worse outcomes across a range of measures, throughout the life course—physical and mental health, education, offending, employment and income—relative to those who have not come in contact with child welfare services.[3]

Reducing the economic burden associated with the consequences of children becoming looked after is of particular concern to policymakers: supporting CLA represents a major expenditure at local authority (LA) level. Across England, between 2011 and 2018, CLA spend increased by £1.9 billion in real terms, to £4.6 billion. Children's services have been described as approaching breaking point.[4] Internationally, there have been increasing calls for a preventive approach to CLA that addresses upstream risk factors for child abuse and neglect.[5]

A number of factors may have contributed to rising rates of children becoming looked after in England over the last decade. High-profile serious case reviews,[6] shifting understanding of the impact of different forms of

## INTRODUCTION

Improving the health outcomes and life chances of Children Looked After (CLA) is a matter of public health concern.[1] In England, over the last decade, the prevalence of CLA increased dramatically, from 54 to 65 per 10 000 children, a rise of 20%. At last count, in March 2019, their number exceeded 78 000.[2] The health outcomes and life chances of these children, many of whom have experienced abuse, neglect and other

childhood adversity[7] and legal judgements clarifying LA statutory responsibilities[8] may all affect thresholds for child welfare intervention. Wider economic changes may also underlie trends in CLA rates. Growing up in adverse socioeconomic circumstances (SECs) is an important risk factor for child abuse and neglect and for children being taken into care,[9] with poverty, unemployment and parental financial stress recognised as contributory causal factors.[10 11] Several experimental and quasi-experimental studies from the USA have shown that raising family income and reducing poverty leads to a reduction in rates of child abuse and neglect.[10 12]

In 2008, the onset of financial recession led to rising unemployment in England and to fiscal policy with far-reaching social consequences. In 2010, the UK government began introducing a series of austerity measures with the stated intention of eliminating the budget deficit and reducing the national debt.[13] The welfare system has been a principal focus of cuts and reforms.[14] These have adversely affected, in particular, families with children and those at greatest risk of poverty, fuelling a rise in child poverty.[15] At the same time, regressive cuts to LA budgets have led to reduced spending on early childhood education and care, and other prevention services.[16] While increases in unemployment during recession were dispersed across all parts of the country, changes in welfare provision and cuts to prevention have disproportionately affected deprived areas.[17] If these changes are leading to increased incidence of child abuse and neglect, we would expect rates of children becoming looked after to rise more rapidly in more deprived areas.

There are stark differences in rates of CLA across LAs in England.[1] Less clear is how these are changing over time. Our aim in this study is to determine whether the rate of children becoming looked after increased more in deprived areas of the country, after controlling for unemployment—so parcelling out the effects of recession itself from the effects of other possible drivers of changing inequalities. We further quantify trends in inequalities in children experiencing other forms of child welfare intervention to assess whether findings for CLA are consistent across child welfare outcomes.

## METHODS
### Data sources and measures
We undertook a longitudinal, local area ecological analysis of rates of children becoming looked after in England. We used routinely available data from 150 upper-tier LAs between 2004 and 2019, based on 2010 boundaries (see online supplemental appendix 1). Two LAs, the City of London and the Isles of Scilly, were excluded due to their small population size.

Our primary outcome of interest was the annual age-standardised rate of children starting to be looked after by LAs in England (hereafter referred to as 'CLA rate'). Panel data for the number of CLA, by age group, were drawn from the 'children looked after data return', submitted by LAs to the Department of Education on 31 March annually.[2] We refer to the financial year by the latter year throughout. Direct age standardisation was performed using the national population distribution of children.

Secondary outcomes captured the wider population of children known to children's social care. Figure 1 outlines the different child welfare outcomes. The system has been likened to a 'funnel', with a progressively smaller number of children experiencing increasingly acute

The children's social care system has been described as series of 'filters and funnels'.[∞] Using this analogy, through the funnel, successive phases of risk assessment and service response determine a child's incident status:

1. **Referrals**. At the wide end of the funnel are all referrals.
2. **Children in Need**. A child is 'in need' (CIN), if deemed to require additional support in order to achieve a reasonable standard of health and development. An episode of need relates to an open case following a LA's acceptance of a referral to children's social care.
3. **Child Protection Plan**. A Child Protection Plan (CPP) may be drawn up where, following an investigation, concerns persist as to whether a child is suffering, or likely to suffer, significant harm.
4. **Children Looked After** (CLA) are at the narrow end of the funnel, enduring adversity sufficiently severe for the State to intervene in their upbringing.[£]

This figure represents the 'funnel' of children's social care, and shows the overlap between incident child welfare status. Size and overlaps are not to scale. Child population estimates are taken from most recent data returns, for the period 2018-19. In rare cases, due to residual safeguarding concerns, a child in care may also be subject to a CPP.

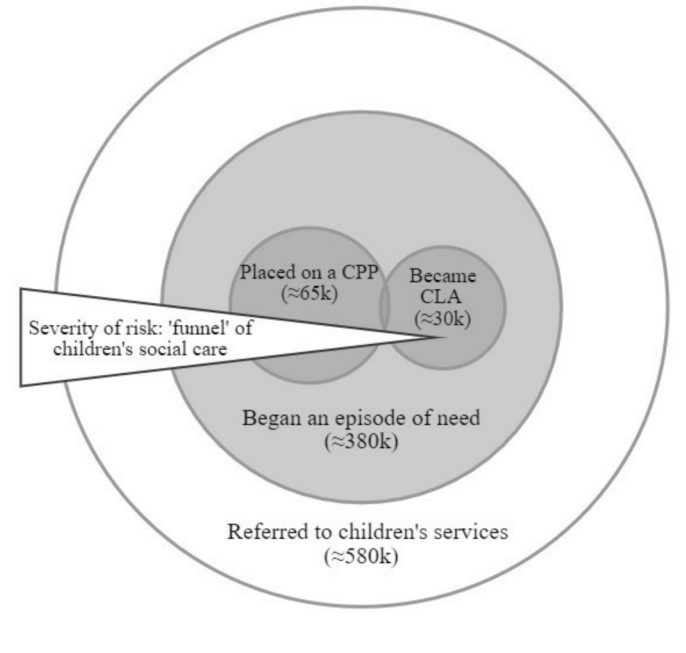

**Figure 1** Description of the children's social care system in England.∞Gibbon et al[18] £Emmott et al[50]

interventions. We used the annual age-standardised rate of children becoming the subject of a Child Protection Plan ('CPP rate') and children beginning an 'episode of need' ('Children in Need (CIN) rate'). Data for these outcomes between 2010 and 2019 were sourced from the CIN census records of children referred for social care support in England.[18] For children on a CPP, a breakdown of numbers by category of abuse was available. Disaggregation by age group was requested via a Freedom of Information request and was obtained for the years 2012–2019.

As a measure of SEC, we used the income deprivation score of the 2010 Indices of Multiple Deprivation.[19] This is a non-overlapping count of individuals who, as a result of low earnings, qualify for means-tested benefits, as a proportion of the total population.[20] We used 2010 scores based on 2008 data, collected prior to the implementation of austerity policies, to avoid conflating the time-invariant measure of deprivation with unmeasured time-varying exposures that may be changing in response to austerity policies, and so contributing to changing inequalities. In descriptive analyses, we categorised the income deprivation score, assigning LAs to quintiles such that 20% of the 2008 child population was apportioned to each quintile. In regression models, we used a continuous measure of the income deprivation score, converted to a weighted rank by assigning a value from 0 to 1 based on the midpoint of the LA's range in the cumulative distribution. When using this value as a continuous exposure variable in the regression model, the estimated coefficient expresses the change in the Slope Index of Inequality (SII), a commonly used indicator of the association between health outcomes and socioeconomic deprivation.[21] The same value can be used to derive the change in the Relative Index of Inequality (RII) when the outcome variable in the regression model is log-transformed and the estimated coefficient exponentiated. In our statistical analyses, the SII represents the absolute difference, and the RII the relative difference, in child welfare outcomes between the LA of lowest and highest level of income deprivation, taking into account the distribution of the child population across LAs.[22]

Our analyses also included LA unemployment rates as a covariate to separate out the impact of recession on child welfare outcomes, and so determine whether changes in inequalities were independent of the effects of unemployment. We used data on the number of people claiming Jobseeker's Allowance, plus those claiming Universal Credit who are out of work, as a proportion of residents aged 16–64 years, in the financial year.[23] Although the measure does not capture all unemployment, it is precise and stable at local area level, is highly correlated with survey-based measures of unemployment[24] and spans the time period of interest. Since the effects of unemployment on child welfare outcomes are unlikely to be immediate, we lagged the variable by 1 year.

## Statistical analysis

First, we assessed descriptive trends for our outcome CLA rate, across LAs grouped into quintiles of income deprivation, between 2004 and 2019. Second, we estimated a segmented linear regression model, with age-standardised CLA rate as the outcome; year, unemployment rate and income deprivation weighted rank as continuous independent variables; and random intercept and slope terms to account for the correlation between measurements within LAs. Based on our initial descriptive analysis, we included a linear spline for the effect of financial year, with one knot indicating the timing of the change in trend. We used an iterative search procedure to confirm the knot position, resulting in the model with the smallest Bayesian information criterion value.[25 26] We included an interaction between the spline terms for the effect of year and deprivation to allow for potential differences in trend by SEC. Full details are provided in online supplemental appendices 2 and 3.

We used this model to assess whether there was a significant change in the trend in CLA rates over this period, whether this differed by level of LA income deprivation and the potential contribution of unemployment to trends in our outcome. We estimated all model parameters by maximum likelihood, using generalised likelihood ratio statistics to compare nested models and Wald statistics to test hypotheses about model parameters. Similar models were fitted for each of our secondary outcomes, CPP and CIN rates, across years for which data were available, 2012–2019—based on our descriptive analysis, no linear splines were included in these models. Models were estimated using the lme4 package[27] in R V.3.5.1. We carried out supplementary analyses, assessing descriptive trends for all outcomes stratified by age and for CPP by category of abuse (see online supplemental appendices 4 and 5) and making predictions based on our main models (see online supplemental appendix 6). Finally, we fit a model with log-transformed values of the age-standardised CLA rate as the outcome to derive the RII and assess trends in relative and absolute inequalities (see online supplemental appendix 7).

## Patient and public involvement

The research question was informed by early conversations with policymakers and practitioners in the Merseyside area and reflects the evidence needs identified by senior leaders within Children's Social Care in a priority-mapping exercise undertaken by the What Works Centre for Children's Social Care.[28] Early plots were shared with local contacts, and the ensuing discussions informed our hypotheses about drivers of recent trends, in particular age-stratified trends. These hypotheses have informed our research agenda.

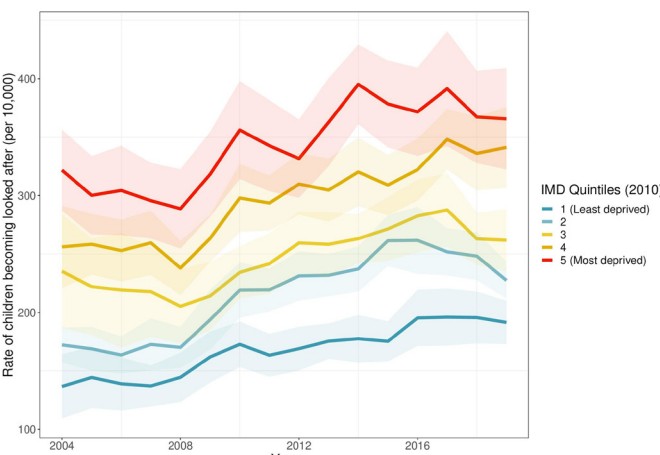

**Figure 2** Children looked after rates by local authority income deprivation quintile, 2004–2019, with 95% CIs. IMD, Index of Multiple Deprivation.

## RESULTS

### Trends in child welfare outcomes

Figure 2 shows CLA rates by LA income deprivation quintile. Between 2004 and 2008, overall CLA rates dipped slightly: a small increase in the most affluent quintile was offset by decreases in more deprived areas. In 2008, the absolute difference in CLA rate between the most and least deprived quintiles was 144 per 100 000 (95% CI 104 to 184). From around 2008, there was a change in trend and an increase in CLA rates. A social gradient in CLA is apparent throughout, with the absolute difference between the most and least deprived quintiles rising to 174 per 100 000 (95% CI 127 to 221) in 2019, an increase of 21% from 2008.

Figure 3 shows CPP and CIN rates by LA income deprivation quintile. As with CLA rates, CPP rates have risen since 2012 and show a clear social gradient. However, the increase occurred relatively evenly across all groups

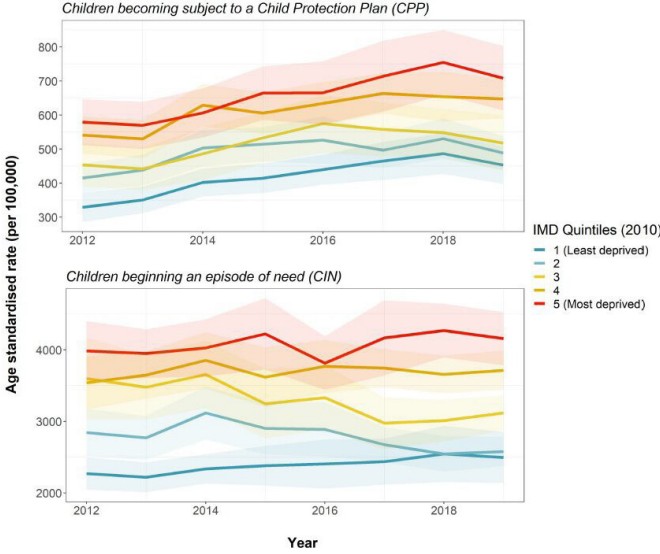

**Figure 3** CPP and CIN rates by local authority income deprivation quintile, 2012–2019, with 95% CIs. IMD, Index of Multiple Deprivation

**Table 1** Association between outcomes and unemployment rate

| Outcome and time period | Annual change (in children per 100 000) for a 1% increase in the unemployment rate the previous year (95% CI) |
| --- | --- |
| CLA rate, 2004–2019 | 9.0 (6.5 to 11.4) |
| CPP rate, 2012–2019 | −10.4 (−22.2 to 1.4) |
| CIN rate, 2012–2019 | 68.5 (−3.1 to 140.1) |

CIN, Children in Need; CPP, Child Protection Plan.

of LAs, in all age groups. CIN rates also exhibit a social gradient, but trends appear to be relatively stable over time.

Supplementary analyses (see online supplemental appendix 4 and figures 1–3) show that the gap in CLA rates between the most and least deprived quintiles differed by age. The gap is wide but relatively stable over time in the youngest age group, children aged under 1 year. The gap is widening in the oldest age group, those aged 16–17 years. Finally, stratifying CPP rates by category of abuse complicates the overall picture of an even rise in rates across all LA income quintiles: we uncovered a widening gap between the most and least deprived quintiles in rates of children becoming subject to a CPP due to concerns about emotional abuse (see online supplemental appendix 5 and figure 4).

### Segmented linear regression models

Tables 1–2 summarise the results of the segmented regression analyses. For full model output and residual diagnostics, see online supplemental appendices 7 and 8 (see online supplemental figures 5–16). For CLA, a knot in 2007, ahead of the 2008 change in trend identified in our descriptive analysis, resulted in the best model

**Table 2** Trends in the slope index of inequality across child welfare outcomes

| Outcome and time period | Annual change (in children per 100 000) in the slope index of inequality (95% CI) |
| --- | --- |
| CLA | |
| 2004–2007 | −11.4 (−22.3 to −0.5) |
| 2007–2019, relative to previous trend | 14.9 (3.6 to 26.2) |
| CPP | |
| 2012–2019 | 4.4 (−11.2 to 20.0) |
| CIN | |
| 2012–2019 | 47.1 (−62.7 to 156.9) |

CIN, Children in Need; CPP, Child Protection Plan.

fit, indicating a change in trend at this point (see online supplemental appendix 3 and figure 17). In our model, rising unemployment in the wake of financial recession was independently associated with rising CLA rates: for each percentage point increase in the unemployment rate, an estimated additional 9 children per 100 000 per year (95% CI 6 to 11) became looked after the following year. There were no associations between CPP and CIN rates and unemployment rates.

But unemployment rates do not account for differences in trends between more and less deprived LAs. In 2004, after controlling for LA unemployment, the SII was 193. This captures the absolute inequality gap across the distribution of LAs on the basis of area deprivation, indicating that there were 193 more CLA per 100 000 in the most deprived LA, compared with the least deprived (95% CI 140 to 246). Between 2004 and 2007, this gap declined by 11 children per 100 000 per year (95% CI 0 to 22) (table 2). From 2007, there was a significant change in the trend in inequalities: the gap increased by 15 children per 100 000 per year (95% CI 4 to 26) relative to the previous trend. Relative inequalities follow the same trend (see online supplemental appendix 7). Altogether, based on our model, we estimate that an additional 18 567 (95% CI 3553 to 33 394) children became looked after between 2007 and 2019, than would have been expected had the rise from 2007 occurred in more deprived LAs as it did in the median LA (see online supplemental appendix 6 and figure 18).

## DISCUSSION
### Main findings
The dramatic rise in CLA in England since 2008 has been greater in poorer areas of the country, increasing inequalities. Overall, an additional 18 567 (95% CI 3553 to 33 394) children started to be looked after between 2007 and 2019 than would have been expected had the rise from 2007 occurred more evenly across LAs. These findings cannot be explained by local economic trends and are consistent with our hypothesis that austerity measures may have contributed to rising rates of child welfare interventions. Our analysis also shows that the rise in CLA was associated with rising unemployment at LA level, a marker of recession.

Trends in inequalities in CLA rates are not simply mirroring broader trends throughout the 'funnel' of children's social care. While CPP rates are also rising, and all outcomes show a clear social gradient, we did not find a greater increase in more deprived compared with less deprived areas for children becoming the subject of a CPP or beginning an episode of need.

Several studies have described trends in child welfare outcomes or child maltreatment in the UK. These support our finding of a change in trend and rising rates from around 2007 to 2008[29] and add context, demonstrating that the turn has followed a 30-year decline in overall rates—although the rise in CPPs due to neglect and

emotional abuse has been occurring since the 1990s.[30] However, to our knowledge, no studies have yet focused on trends in inequalities in child welfare outcomes. Paul Bywaters and colleagues[1] at the Child Welfare Inequalities Project began producing evidence of persistent and systematic inequalities in child welfare outcomes in the UK beginning in 2015. This longitudinal analysis of inequalities is indebted to their work.

### Strengths and limitations
This study is the first to quantify inequalities in child welfare outcomes longitudinally. A strength is that it uses routinely available data for the whole of England and explores several child welfare outcomes to describe trends throughout the child welfare system.

There are several important study limitations. Due to the lack of individual-level data, we used an ecological area-level analysis and cannot identify whether children becoming looked after were directly affected by income deprivation and unemployment. Conceptually, our portrayal of children's social care as a funnel reflects a theoretical model of how a well-functioning system might operate (figure 1) and may not reflect the trajectory of many individual children and families experiencing child welfare intervention. The association between income deprivation and unemployment rates and child welfare outcomes in our analysis may be due to trends in unobserved time-varying confounding factors that varied between LAs.

Trends in the data reflect the interaction between underlying need and response of children's services, and we interpret our findings in this light, with caution. Previous analyses by Bywaters and colleagues demonstrated the existence of an 'inverse intervention law' in child welfare outcomes: a greater risk of intervention in affluent compared with deprived LAs for the same level of neighbourhood deprivation, despite lower overall intervention rates. Several possible explanations relate to supply-side factors, culminating in higher thresholds in more deprived areas.[31] Our models at the level of LAs do not account for the inverse intervention law. However, in combination with reports of rising thresholds due to the rationing of services in more resource-constrained settings, the inverse intervention law[32] may add weight to our findings. Insofar as they reflect changing underlying need, our estimates of the SII are likely to be highly conservative.

### Potential explanations of our findings
#### Changing practice
Several changes in practice during this time period may have influenced thresholds for intervention. First, the death by violence of baby Peter Connelly occurred in 2007, when we see a change in the trend in CLA rates in our data.[33] Media and political narratives that emerged in the aftermath of his death centred on the failure of children's services to intervene,[34] and ensuing reports by The Children and Family Court Advisory and Support

Service note a 'Baby P effect', a marked short-term rise in applications for care orders in a risk-averse environment.[6] This likely accounts for some of the changes in trend and initial rise in CLA rates from 2007. Others have argued that a greater policy focus on early intervention and adoption to improve outcomes for children experiencing adversity has led to a more interventionist, less family-oriented approach.[35] Second, in 2009 the Southwark Judgement clarified and reinforced LAs' statutory duties in relation to those aged 16–17 years presenting to the LA as homeless.[8] This, together with a general shift in practice towards regarding adolescents as vulnerable children rather than nascent adults[36] and greater awareness of extrafamilial forms of abuse and principles of contextual safeguarding,[37] may be contributing to the rising rates among those aged 16–17 years, across all outcomes. However, these phenomena are unlikely to fully explain the long-term rise in CLA rates disproportionately affecting more deprived areas.

### Economic trends

We found evidence of a positive association between unemployment and CLA rates. Although evidence from the UK is scarce, this aligns with Gillham *et al*'s finding of a correlation between male unemployment and child physical abuse in Scotland in the early 1990s[38] and more recent and extensive evidence from the USA demonstrating an association between recession and increased risk of abuse.[39–41] The family stress model posits that heightened stress due to adverse SECs may erode mental health and strain domestic relationships, leading to negative parenting behaviours and increased risk of child abuse and neglect.[42] Barr *et al*'s study of the mental health impact of recession lends credence to this theorised mechanism, demonstrating an association between unemployment and mental health problems in the UK over the same period.[17] Yet unemployment did not fully explain changes in CLA rates in our analysis, and unemployment rates fell rapidly between 2012 and 2019: unemployment cannot explain the continued increase in CLA rates beyond 2012, nor does it explain rising inequalities. Austerity policies subsequent to the initial recession 'shock' may have compounded poor outcomes, affecting inequalities in CLA in several ways.

### Changes to welfare provision and prevention

Regressive cuts to English LA budgets, with deeper cuts in more deprived areas, have precipitated a shift in expenditure away from prevention towards acute services.[16] Between 2011 and 2018, spending on CLA increased by 68% in real terms, whereas spending on early years preventive services (including Sure Start) and non-statutory young people's services fell about 21%. Reports of rising thresholds for intervention in more resource-constrained settings, particularly for early help, have raised concerns that we are 'storing up trouble' for the future.[32] Rationing of early help services may help explain both the relatively stable trends in the less acute CIN rates, and the surge in

children becoming looked after who might have benefited from early support, greater in more deprived LAs. Adolescents may be particularly susceptible to the consequences of austerity, exposed as they are on multiple fronts, not just in the household and schools but increasingly in the wider community. Combined cuts to welfare benefits, youth services,[43] children's mental health services[44] and community policing[45] might disproportionately affect adolescents in more deprived areas, contributing to widening inequalities in this age group.

Changes to welfare benefits have led to rising child poverty, a contributory causal factor in child abuse and neglect.[10 15] Averages losses in earning were particularly high in the more deprived West Midlands and the North West.[15] The most vulnerable children on the edge of care, living in families already struggling to cope, may be particularly sensitive to changes in welfare benefit provision. In particular, the phased introduction of Universal Credit from 2013, with its monthly payments in arrears, enhanced conditionality and punitive sanctions, may have compounded financial stress[15] and parental mental health.[46] This may have increasingly led to more children becoming looked after in deprived areas, contributing to trends in inequalities uncovered in our study. Further research is needed to investigate the impact of changing LA prevention spend and child poverty on child welfare outcomes.

### Policy and practice implications

We demonstrate that the increase in CLA rates from 2007 has been greater in more deprived LAs. Although it is not possible to say what constitutes an appropriate CLA rate,[47] a differential rise by LA deprivation that cannot be fully explained by recession or changing practice is consistent with an increase in underlying need fuelled by welfare changes and cuts to prevention services. While antipoverty social work practice has a crucial role to play in safely reducing CLA rates and inequalities,[48] this must be supported by wider policies to address the social conditions of children's lives. At the national level, this must begin with a renewed commitment to ending child poverty. Expanded social security for families with children and increased funding for LA children's services are safeguarding priorities. LAs' provision of early help services should be placed on a stricter statutory footing, following recommendation 10 of the 2011 Munro Review of Child Protection.[49] In the meantime, at the local level, holding the line on prevention services, amidst statutory pressures, may yield long-term social and economic benefits. Investment in children is key.

**Contributors** DLB is lead author and guarantor. DT-R and BB are joint senior author. DLB and DT-R planned the study. DLB, BB and DT-R analysed the data, supported by KEM, DKS, SW, ETCL and AA. DLB, DT-R and BB led the drafting and revision of the manuscript. All authors contributed to the interpretation of the data and revision of the manuscript; all authors approved the submitted version of the manuscript.

**Funding** DB, DT-R, BB, KEM and AA are funded by the National Institute for Health Research School for Public Health Research (SPHR), Grant Reference

Number PD-SPH-2015. SW is funded by a Wellcome Trust Society and Ethics Research Fellowship (200335/Z/15/Z). DT-R and ETCL are funded by the Medical Research Council (MRC) on a Clinician Scientist Fellowship (MR/P008577/1). BB is also supported by the National Institute for Health Research Applied Research Collaboration North West Coast (ARC NWC). The views expressed in this publication are those of the authors and not necessarily those of the National Institute for Health Research or the Department of Health and Social Care, MRC or Wellcome Trust. The funders of the study had no role in study design, data collection, data analysis, data interpretation or the writing of the report. The corresponding author had access to all of the data in the study and had final responsibility for the decision to submit for publication.

**Competing interests** None declared.

**Patient consent for publication** Not required.

**Ethics approval** Research ethics committee approval was not required for this secondary data analysis, which employed publicly available aggregate data.

**Provenance and peer review** Not commissioned; externally peer reviewed.

**Data availability statement** Data are available in a public, open access repository. The publicly available, aggregate data can be accessed from the Place-based Longitudinal Data Resource (PLDR, https://pldr.org/), and the code is available upon reasonable request.

**ORCID iDs**
Davara Lee Bennett http://orcid.org/0000-0003-3480-6566
Kate E Mason http://orcid.org/0000-0001-5020-5256

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
