## [Reviewer comments · BMJ Open]

ARTICLE DETAILS

TITLE (PROVISIONAL)	Trends in inequalities in looked after children in England 2004-2019: a local area ecological analysis
AUTHORS	Bennett, Davara Lee; Mason, Kate E; Schlüter, Daniela K; Wickham, S; Lai, Eric TC; Alexiou, Alexandros; Barr, Ben; Taylor-Robinson, David

VERSION 1 - REVIEW

REVIEWER	Julian Gardiner University of Oxford, UK
REVIEW RETURNED	26-Jun-2020

GENERAL COMMENTS	This paper addresses an important question, using a comprehensive data source. The methodology is sound and the models used are clearly presented. Overall, the paper is well-written and easy to follow. This paper addresses an important question, using a comprehensive data source. The methodology is sound and the models used are clearly presented. Overall, the paper is well-written and easy to follow. I have only very minor suggestions for corrections / clarifications. Page 7, lines 37-41 “Since the effect of unemployment on child welfare outcomes are unlikely to be immediate, we lagged the variable by one year.” Suggest change to: “Since the effects of unemployment on child welfare outcomes are unlikely to be immediate, we lagged the variable by one year.” Page 10, lines 50-51 Here “funnel” is mentioned for the first time. Suggest it would be helpful to explain the “funnel of care” here, or at least give a reference to Figure 1. Page 12, lines 56-57 “contributing to a widening inequalities”
--

	Suggest change to: “contributing to widening inequalities” Page 19, Figure 1 In the draft, the text towards the right-hand side of this figure is too small to read. I hope it will be possible to make this figure larger in the published version. Pages 25-26 The note at the bottom of the three tables on these pages reads: *p**p***p<0.01 This needs to be corrected in some way. Page 32 The legend to the right of the figure reads in part: Predicted Expected What is meant, I think, is: Predicted Observed
--	---

REVIEWER	David Walsh Glasgow Centre for Population Health, Scotland
REVIEW RETURNED	09-Jul-2020

GENERAL COMMENTS	I found this to be a well written, generally very clearly explained, paper on what is a supremely important topic - trends in looked after children. The authors should be congratulated on what has clearly been a detailed, in-depth set of analyses (I always enjoy reviewing a paper with seven appendices..). Despite what I have just said about the clear explanation in the paper, my main (and NB only) major issue with it relates to clarity of message. The abstract makes it clear that you are examining the impact of the recession on looked after children. Obviously, however, what has been more detrimental to a range of social and health outcomes and inequalities in the UK has not just been the recession, but the political response to it i.e. a decade of so-called austerity. The authors know this because they very eloquently state its importance in both the introduction and discussion - indeed in the latter it is (I think correctly) very much proposed as an important part of the explanation for the study's main findings. But that obviously begs the question of why they have framed the study solely around the recession (with unemployment as a marker of it) instead of – or as well as – looking at markers of austerity? A number of variables could have been used for the latter – and again the authors know this because some of them have used such data in other published analyses. I'm left with the sense (and this is probably unfair) that
--

	they perhaps wanted to frame this around austerity, but because they haven't or couldn't, they have done so vis-à-vis the recession instead. I'm sure that's not the case, but it's the sense suggested. But it's really, really important that the message is clear: if the driver of the trends is political decision-making (rather than a recession caused by other global factors), it's important that that is clearly stated, including in the abstract as well (because, depressingly, that's the only part of the paper most people read). (Related to the above, unemployment is not a great variable anyway (especially when based on claimant counts rather than some kind of ILO-defined survey measure): could you not have used a child poverty rate which might have been more relevant to the subject matter, and also might better capture some of the other causal factors you are interested in?) An ideal response to this would be to include a marker of austerity in the analyses. If that's not possible, then the issue outlined above at least needs to be addressed in the Discussion. The latter section already states that both the recession and austerity are likely be drivers – you should at least add a justification there for why you've only looked at one of those two factors. The other points are all fairly minor, viz.:  • In terms of context, the Discussion doesn't say very much about how the findings compare with those of other relevant research – which would obviously be expected in this type of paper. I suspect that's just because there hasn't been a great deal done other than what you quote (e.g. a Scottish analysis from the 1990s). But if that's the case, it would be helpful just to say so; and if it's not the case (i.e. you haven't discussed other relevant studies) you should obviously do so. There's also a broader issue here of contextualising the findings i.e. had inequalities been widening before the period of your analysis? That kind of context is quite important to know and understand. • You looked at absolute inequalities rather than relative (or indeed both): it would be helpful to state why you think that's more appropriate • Figure 1 is really difficult to read in the printed version of the paper – can it be made bigger?
--	---

VERSION 1 – AUTHOR RESPONSE

Response to Reviewer(s)' Comments

Reviewer #1 (R1):

R1: This paper addresses an important question, using a comprehensive data source. The methodology is sound and the models used are clearly presented. Overall, the paper is well-written and easy to follow. I have only very minor suggestions for corrections / clarifications.

Page 7, lines 37-41

“Since the effect of unemployment on child welfare outcomes are unlikely to be immediate, we lagged the variable by one year.”

Suggest change to:

“Since the effects of unemployment on child welfare outcomes are unlikely to be immediate, we lagged the variable by one year.”

Response: Thank you for your positive feedback on our article, and for reading so carefully. We have corrected the typographical error.

R1: Page 10, lines 50-51

Here “funnel” is mentioned for the first time. Suggest it would be helpful to explain the “funnel of care” here, or at least give a reference to Figure 1.

Response: Thank you. We have added an explanation of the ‘funnel’ earlier in the paper, when figure 1 is first introduced, so that subsequent references will be understood. We have also referenced figure 1 where it will not distract from the interpretation of results.

Page 6 lines 51-52

The system has been likened to a ‘funnel’, with a progressively smaller number of children experiencing increasingly acute interventions.

Page 11 lines 12-13

Conceptually, our portrayal of children’s social care as a funnel reflects a theoretical model of how a well-functioning system might operate (figure 1), and may not reflect the trajectory of many individual children and families experiencing child welfare intervention.

R1: Page 12, lines 56-57

“contributing to a widening inequalities”

Suggest change to:

“contributing to widening inequalities”

Response: Thank you, well caught, and amended in the revised manuscript.

R1: Page 19, Figure 1

In the draft, the text towards the right-hand side of this figure is too small to read. I hope it will be possible to make this figure larger in the published version.

Response: Thank you for pointing this out. We have amended the figure, increasing the font size by several points, and giving more space to the figure in the textbox.

R1: Pages 25-26

The note at the bottom of the three tables on these pages reads:

*p**p***p<0.01

This needs to be corrected in some way.

Response: Thank you, this was indeed unclear. We have consolidated the model output in a single table. At the same time, we have amended the note beneath each of the models so that it now reads as follows:

* p < 0.05, ** p < 0.01

Accordingly, where one or multiple asterisks appear, we have removed one asterisk to bring it in line with the legend.

R1:

Page 32

The legend to the right of the figure reads in part:

Predicted

Expected

What is meant, I think, is:

Predicted

Observed

Response: Thank you for this and apologies for the confusion. In appendix figure 2 we were comparing descriptive trends derived from our model (labelled 'Predictive', for consistency with the label for the same trend in appendix figure 9), with hypothetical trends derived from our model, but based on a scenario in which the rise in CLA rates occurred more evenly across areas (previously labelled 'Expected').

We agree that this was confusing. We have therefore amended the plot so that we are comparing real trends (labelled 'Observed', for consistency with the same trend in appendix figure 9), with hypothetical trends derived from our model, and based on the same scenario in which the rise in CLA rates occurred more evenly across areas (now labelled 'Predicted').

We have amended the legend, using the labels proposed:

Observed

Predicted

We have correspondingly amended appendix 6 to reflect this language:

Appendix figure 12, showing LAs grouped by quintiles, illustrates predicted rates according to this scenario.

Reviewer #2 (R2):

R2: I found this to be a well written, generally very clearly explained, paper on what is a supremely important topic - trends in looked after children. The authors should be congratulated on what has clearly been a detailed, in-depth set of analyses (I always enjoy reviewing a paper with seven appendices...)

Response: Thank you very much for your encouraging comments, and we are glad that you enjoyed the appendices – though we have tried to avoid their further proliferation.

Despite what I have just said about the clear explanation in the paper, my main (and NB only) major issue with it relates to clarity of message. The abstract makes it clear that you are examining the impact of the recession on looked after children. Obviously, however, what has been more detrimental to a range of social and health outcomes and inequalities in the UK has not just been the recession, but the political response to it i.e. a decade of so-called austerity. The authors know this because they very eloquently state its importance in both the introduction and discussion - indeed in the latter it is (I think correctly) very much proposed as an important part of the explanation for the study's main findings. But that obviously begs the question of why they have framed the study solely around the recession (with unemployment as a marker of it) instead of – or as well as – looking at markers of austerity? A number of variables could have been used for the latter – and again the authors know this because some of them have used such data in other published analyses. I'm left with the sense (and this is probably unfair) that they perhaps wanted to frame this around austerity,

but because they haven't or couldn't, they have done so vis-à-vis the recession instead. I'm sure that's not the case, but it's the sense suggested. But it's really, really important that the message is clear: if the driver of the trends is political decision-making (rather than a recession caused by other global factors), it's important that that is clearly stated, including in the abstract as well (because, depressingly, that's the only part of the paper most people read).

(...)

An ideal response to this would be to include a marker of austerity in the analyses. If that's not possible, then the issue outlined above at least needs to be addressed in the Discussion. The latter section already states that both the recession and austerity are likely be drivers – you should at least add a justification there for why you've only looked at one of those two factors.

Response: Thank you for the astute comment. We agree that the question of the impact of austerity policies on looked after children is an incredibly important one. However, addressing this properly is beyond the scope of this particular paper, which focuses primarily on trends in inequalities across several child welfare outcomes after controlling for the impact of the recession, and across different age groups and categories of abuse. Although we do not feel that there is space to include further analyses in this paper, we will go on to address the potential drivers of rising inequalities (child poverty and prevention spend) in a subsequent paper, using different statistical methods.

We agree that the rationale for including unemployment in the model was not sufficiently clear, and hope that the following changes will have clarified and strengthened the main message.

p.2: Objective: To determine whether there were inequalities in the precipitous rise in rates of children becoming looked after (CLA) in England over the last decade, and the contribution of unemployment, a marker of the recession, to these trends.

p. 2: Slope indices of inequality (SII) were estimated using longitudinal segmented mixed effects models, controlling for unemployment, a marker of the recession.

p.2: Between 2007 and 2019 the gap between most and least deprived areas increased by 15 children per 100,000 per year (95% CI 4-26) relative to the previous trend, and after controlling for local economic trends.

p.5.: Our aim in this study is to determine whether the rate of children becoming looked after increased more in deprived areas of the country, and the contribution of unemployment, a marker of recession, to these trends.

p. 6.: We used 2010 scores based on 2008 data, collected prior to the implementation of austerity policies, to avoid conflating the time-invariant measure of deprivation with unmeasured time-varying exposures that may be changing in response to austerity policies, and so contributing to changing inequalities.

p.6.: In descriptive analyses, we categorised the income deprivation score, assigning LAs to quintiles such that 20% of the 2008 child population was apportioned to each quintile.

p.6.: In regression models, we used a continuous measure of the income deprivation score, converted to a weighted rank by assigning a value from 0 to 1 based on the midpoint of the LA's range in the cumulative distribution.

p. 6: Our analyses also included LA unemployment rates as a covariate in order to separate out the impact of the recession on child welfare outcomes, and so determine whether changes in inequalities were independent of the effects of unemployment.

p. 9: These findings cannot be explained by local economic trends, and are consistent with our hypothesis that austerity measures may have contributed to rising rates of child welfare outcomes.

p.11: Further research is needed to investigate the impact of changing LA prevention spend and child poverty on child welfare outcomes. [NB: this sentence has been moved to the end of the section to summarise the need for research into both these variables as a next step].

p.12: Although it is not possible to say what constitutes an appropriate CLA rate (40), a differential rise by LA deprivation that cannot be explained by the recession is consistent with an increase in underlying need fuelled by welfare changes and cuts to prevention services.

R2: Related to the above, unemployment is not a great variable anyway (especially when based on claimant counts rather than some kind of ILO-defined survey measure): could you not have used a child poverty rate which might have been more relevant to the subject matter, and also might better capture some of the other causal factors you are interested in?

Response: Thank you for this comment. Our response to the previous comment aims to address the reasons for controlling for unemployment, rather than focussing on the effect of child poverty or other austerity-related measures. With regards to data sources, there were a few reasons why we considered claimant count data to be preferable to survey-based measures of unemployment in this instance. It has been used in a number of other papers on the impact of the recession and may be a more precise and stable measure than survey-based measures at small area level. In their analysis of suicides associated with the recession, Barr et al also noted that the two measures are highly correlated. Finally, the Annual Population Survey data is only available from calendar year 2004, whereas complete data on Looked After Children are available from financial year 2003-04. If we wished to ensure that the exposure occurred before the outcome, lagging by a year, we would lose two years' worth of valuable information about trends in inequalities in Looked After Children. Given the main focus of the paper, we feel that the use of claimant count data is justified.

That said, we are glad of the opportunity to acknowledge the limitations of the data while justifying its use. We have added the following sentence and reference:

p.6: Although the measure does not capture all unemployment, it is precise and stable at local-area level, is highly correlated with survey-based measures of unemployment (24), and spans the time period of interest.

(24) Barr B, Taylor-Robinson D, Scott-Samuel A, McKee M, Stuckler D. Suicides associated with the 2008-10 economic recession in England: Time trend analysis. *BMJ (Online)* [Internet]. 2012 Sep 8 [cited 2020 Jul 30];345(7873). Available from: <https://www.bmj.com/content/345/bmj.e5142>

R2: The other points are all fairly minor, viz.:

In terms of context, the Discussion doesn't say very much about how the findings compare with those of other relevant research – which would obviously be expected in this type of paper. I suspect that's just because there hasn't been a great deal done other than what you quote (e.g. a Scottish analysis from the 1990s). But if that's the case, it would be helpful just to say so; and if it's not the case (i.e. you haven't discussed other relevant studies) you should obviously do so.

Response: Thank you for your comment. In the Discussion section of the revised manuscript, we have engaged more fully with the literature on: trends in child welfare outcomes (referencing several paper); thresholds for intervention (adding a reference); and trends in inequalities in child welfare outcomes (noting the lack of research in this area). We have also reworded a sentence about associations between recession effects and child maltreatment, noting the scarcity of research in the UK, but adding several references to further studies from the US.

p.10: Several studies have described trends in child welfare outcomes or child maltreatment in the UK. These support our finding of a change in trend and rising rates from around 2007-08 (29) and add context, demonstrating that the turn has followed a thirty-year decline in overall rates – though the rise in CPPs due to neglect and emotional abuse have been rising since the 1990s (30). However, to our knowledge no studies have yet documented trends in inequalities. Paul Bywaters and colleagues at the Child Welfare Inequalities Project began producing evidence of persistent and systematic inequalities in child welfare outcomes in the UK beginning in 2015 (3). This longitudinal analysis of inequalities is indebted to their work.

29. Chandan JS, Gokhale KM, Bradbury-Jones C, Nirantharakumar K, Bandyopadhyay S, Taylor J. Exploration of trends in the incidence and prevalence of childhood maltreatment and domestic abuse recording in UK primary care: a retrospective cohort study using “the health improvement network” database. *BMJ open*. 2020;

30. Degli Esposti M, Humphreys DK, Jenkins BM, Gasparrini A, Pooley S, Eisner M, et al. Long-term trends in child maltreatment in England and Wales, 1858–2016: an observational, time-series analysis. *The Lancet Public Health*. 2019 Mar 1;4(3):e148–58.

p.11: Others have argued that a greater policy focus on early intervention and adoption in order to improve outcomes for children experiencing adversity has led to a more interventionist, less family-oriented approach (34).

34. Featherstone B, Morris K, White S. A marriage made in hell: Early intervention meets child protection. *British Journal of Social Work* [Internet]. 2014 [cited 2020 Jun 19];44(7):1735–49. Available from: <https://academic.oup.com/bjsw/article-abstract/44/7/1735/1718126>

p.11-12: Though evidence from the UK is scarce, this aligns with Gillham et al.’s finding of a correlation between male unemployment and child physical abuse in Scotland in the early 1990s (37) and more recent and extensive evidence from the US demonstrating an association between the recession and increased risk of abuse (38–40).

39. Cherry R, Wang C. The link between male employment and child maltreatment in the U.S., 2000-2012. *Children and Youth Services Review*. 2016;66:117–22.

40. Millett L, Lanier P, Drake B. Are economic trends associated with child maltreatment? Preliminary results from the recent recession using state level data. *Children and Youth Services Review*. 2011 Jul 1;33(7):1280–7.

R2: There’s also a broader issue here of contextualising the findings i.e. had inequalities been widening before the period of your analysis? That kind of context is quite important to know and understand.

Response: Thank you. We hope that the paragraph we added in response to the last comment also addresses this query. To our knowledge, there have been no studies quantifying inequalities in children entering care longitudinally, and so we do not know how inequalities were changing prior to the study period. However, there is evidence of a thirty-year decline in rates of children entering care

prior to 2004 – though long-term trends in emotional abuse and neglect seem to have been pulling against this decline.

R2: You looked at absolute inequalities rather than relative (or indeed both): it would be helpful to state why you think that's more appropriate

Response: Thank you for this comment. We had chosen to present absolute inequalities to maximise the policy relevance of the findings. Presenting the changing gap between most and least deprived areas in terms of absolute numbers of children helps contextualise the findings. However, we believe that both absolute and relative inequalities are important, and may be of interest to some. To that end, we have additionally presented exponentiated results of the regression model with log-transformed CLA rates as the outcome. Interpreting the RII, we can see the same picture: as with absolute inequalities, relative inequalities declined to 2007, and rose thereafter, relative to previous trends. Therefore, for the sake of brevity and simplicity, we have discursively interpreted only absolute inequalities in the body of the manuscript, presenting the output for the additional analysis in appendix 7.

Changes to the manuscript are as follows:

p.6: The same value can be used to derive the change in the Relative Index of Inequality (RII) when values of the outcome variable in the regression model are log-transformed. In our statistical analyses, the SII represents the absolute difference, and the RII the relative difference, in child welfare outcomes between LAs of lowest and highest levels of income deprivation, taking into account the distribution of the child population across LAs (22).

p.7: Finally, we fit a model with log-transformed values of the age standardised CLA rate as the outcome in order to derive the RII, and assess trends in relative, as well as absolute inequalities (appendix 7).

p.8: Relative inequalities follow the same trend (appendix 7).

Changes to the appendices are as follows:

p.3: The following tables summarise the full output for each of the models in turn:

- Age standardised CLA rates
- Age standardised CLA rates, log-transformed (results exponentiated)
- Age standardised CPP rates

p.4: Table, showing the model output for this additional model.

p. 5:

- a. CLA model (absolute inequalities): see appendix figures 5-7
- b. CLA model (relative inequalities): see appendix figures 8-10

R2: Figure 1 is really difficult to read in the printed version of the paper – can it be made bigger?

Response: Yes, thank you for pointing this out. We have amended the figure, increasing the font size by several points, and giving more space to the figure in the textbox.

VERSION 2 – REVIEW

REVIEWER	David Walsh Glasgow Centre for Population Health, Scotland
REVIEW RETURNED	24-Aug-2020

GENERAL COMMENTS	Thank you for making the changes in relation to the previous comments. I really don't want to drag this out, as it's obviously an excellent, detailed, set of analyses on a hugely important topic, and the results are hugely important too – but I think it still needs a bit of editing, principally (but not exclusively) to the abstract. The changes made to the abstract have been minimal, and I really don't think it properly captures some of the key results and implications of the paper. It doesn't explain why you're controlling for unemployment, it completely omits the key finding that unemployment doesn't explain the widening of inequalities, and therefore it also omits the implications of that i.e. what you say in the paper about likely other causes. So without me writing it for you (!), it needs to say (1) what the aim was; (2) why – as part of the aim – you control for unemployment (i.e. because of the association between poverty/socioeconomic conditions and maltreatment etc); (3) that rates increased over time; (4) that inequalities widened; (5) that unemployment was associated with the increase; (5) but really importantly, that unemployment did not explain the widening inequalities; (6) therefore you hypothesise that other factors (i.e. austerity) are the most likely explanation. You obviously include some of that but not all of it, and it's really not as clear as it should be. Other required changes: Also in the abstract: 'controlling for local economic trends' is just a bit confusing: just say unemployment – that's what it is, and what you've already stated you're controlling for (unless it's referring to something else?!) Still in the abstract, 'relative to the previous trend' doesn't make much sense here. State the years of the previous period you're talking about. You also need to clarify the aims in the main part of the paper too. It's still got a slightly muddled hangover from the first version i.e. it mentions recession and austerity in the build-up to stating the aims of the study, but then says nothing about either when the aims are stated. So again it needs to add a bit re. controlling for unemployment and why you're doing that (as per the abstract). Typo in 2nd paragraph of methods ('CLA' should be 'CLAs') These changes will take you about five minutes to make, but I think they will make the paper (or at least the abstract) much clearer.
---

	Other than that, well done on producing such an important set of analyses, and I look forward to seeing the paper published.
--	--

VERSION 2 – AUTHOR RESPONSE

Response to Reviewer's Comments

R2: Thank you for making the changes in relation to the previous comments. I really don't want to drag this out, as it's obviously an excellent, detailed, set of analyses on a hugely important topic, and the results are hugely important too – but I think it still needs a bit of editing, principally (but not exclusively) to the abstract. The changes made to the abstract have been minimal, and I really don't think it properly captures some of the key results and implications of the paper. It doesn't explain why you're controlling for unemployment, it completely omits the key finding that unemployment doesn't explain the widening of inequalities, and therefore it also omits the implications of that i.e. what you say in the paper about likely other causes.

So without me writing it for you (!), it needs to say (1) what the aim was; (2) why – as part of the aim – you control for unemployment (i.e. because of the association between poverty/socioeconomic conditions and maltreatment etc); (3) that rates increased over time; (4) that inequalities widened; (5) that unemployment was associated with the increase; (6) but really importantly, that unemployment did not explain the widening inequalities; (6) therefore you hypothesise that other factors (i.e. austerity) are the most likely explanation. You obviously include some of that but not all of it, and it's really not as clear as it should be.

Response: Thank you. We fully agree with the points you raise and really appreciate the opportunity to iron out the remaining issues. We have rewritten the abstract, incorporating all suggestions:

p. 2:

Objective: To assess trends in inequalities in children becoming looked after (CLA) in England between 2004 and 2019, after controlling for unemployment, a marker of the recession and risk factor for child maltreatment.

Design: longitudinal local area ecological analysis

Setting: 150 English upper-tier local authorities.

Participants: Children under the age of 18.

Primary outcome measure: The annual age-standardised rate of children becoming looked after (CLA rate) across English local authorities, grouped into quintiles based on their level of income deprivation. Slope indices of inequality (SII) were estimated using longitudinal segmented mixed effects models, controlling for unemployment.

Results: Since 2008, there has been a precipitous rise in CLA rates, and a marked widening of inequalities. Unemployment was associated with rising CLA rates: for each percentage point increase in the unemployment rate, an estimated additional 9 children per 100,000 (95% CI 6-11) per year became looked after the following year. However, inequalities increased independently of the effect of unemployment. Between 2007 and 2019, after controlling for unemployment, the gap between most and least deprived areas increased by 15 children per 100,000 per year (95% CI 4-26) relative to the 2004-2006 trend.

Conclusions: The dramatic increase in the number of CLA has been greater in poorer areas, and in areas more deeply affected by the recession. But trends in unemployment do not explain the decade-long rise in inequalities, suggesting that other socioeconomic factors, including rising child poverty and reduced spending on Children's Services, may be fuelling inequalities. Policies to safely reduce the rate of children becoming looked after should urgently address the social determinants of child health and wellbeing.

R2: Also in the abstract: 'controlling for local economic trends' is just a bit confusing: just say unemployment – that's what it is, and what you've already stated you're controlling for (unless it's referring to something else?!)

Response: We agree that the terminology should be more precise. We are referring here to unemployment, and have amended as follows:

Between 2007 and 2019, after controlling for unemployment, the gap between most and least deprived areas increased by 15 children per 100,000 per year (95% CI 4-26) relative to the 2004-2006 trend, and after controlling for local economic trends.

R2: Still in the abstract, 'relative to the previous trend' doesn't make much sense here. State the years of the previous period you're talking about.

Response: Thank you and agreed. We have now included this information:

Between 2007 and 2019, after controlling for unemployment, the gap between most and least deprived areas increased by 15 children per 100,000 per year (95% CI 4-26) relative to the previous 2004-2006 trend.

R2: You also need to clarify the aims in the main part of the paper too. It's still got a slightly muddled hangover from the first version i.e. it mentions recession and austerity in the build-up to stating the aims of the study, but then says nothing about either when the aims are stated. So again it needs to add a bit re. controlling for unemployment and why you're doing that (as per the abstract).

Response: Thank you – we agree and have added a clause, clarifying the logic:

p. 6: "Our aim in this study is to determine whether the rate of children becoming looked after increased more in deprived areas of the country, after controlling for unemployment – so parcelling out the effects of the recession itself from the effects of other possible drivers of changing inequalities."

R2: Typo in 2nd paragraph of methods ('CLA' should be 'CLAs')

Response: Thank you. We acknowledge that this may read as a typo, though in this case we did make a choice. We have preferred to use 'CLA' ('Children Looked After') as an acronym already in the plural, rather than 'LAC' ('Looked After Child'). It is a small concession to the preferences of many children and young people who, it is reported, dislike the term when applied to themselves as individuals. The NSPCC reports that many children prefer the term 'in care', but this may cause confusion due to its specific legal meaning. We are happy to make the change if it is an editorial preference, but wished to first outline our reasoning. See for example: <https://learning.nspcc.org.uk/children-and-families-at-risk/looked-after-children#:~:text=A%20child%20who%20has%20been,children%20and%20young%20people%20prefer.>

<https://www.calderdale.gov.uk/v2/residents/health-and-social-care/joint-strategic-needs-assessment/children-and-young-people/cla-health>

R2: These changes will take you about five minutes to make, but I think they will make the paper (or at least the abstract) much clearer. Other than that, well done on producing such an important set of analyses, and I look forward to seeing the paper published.

Response: Thank you for this and your kind words. We agree that these final touches have vastly improved the paper. It is easy to unintentionally obscure the thrust of the argument when the clarity of expression is lacking.